# Application of Copper Nanoparticles in Dentistry

**DOI:** 10.3390/nano12050805

**Published:** 2022-02-27

**Authors:** Veena Wenqing Xu, Mohammed Zahedul Islam Nizami, Iris Xiaoxue Yin, Ollie Yiru Yu, Christie Ying Kei Lung, Chun Hung Chu

**Affiliations:** Faculty of Dentistry, University of Hong Kong, Hong Kong 999077, China; u3008489@connect.hku.hk (V.W.X.); irisxyin@hku.hk (I.X.Y.); ollieyu@hku.hk (O.Y.Y.); yklung@graduate.hku.hk (C.Y.K.L.); chchu@hku.hk (C.H.C.)

**Keywords:** copper nanoparticles, antimicrobial, dentistry

## Abstract

Nanoparticles based on metal and metallic oxides have become a novel trend for dental applications. Metal nanoparticles are commonly used in dentistry for their exclusive shape-dependent properties, including their variable nano-sizes and forms, unique distribution, and large surface-area-to-volume ratio. These properties enhance the bio-physio-chemical functionalization, antimicrobial activity, and biocompatibility of the nanoparticles. Copper is an earth-abundant inexpensive metal, and its nanoparticle synthesis is cost effective. Copper nanoparticles readily intermix and bind with other metals, ceramics, and polymers, and they exhibit physiochemical stability in the compounds. Hence, copper nanoparticles are among the commonly used metal nanoparticles in dentistry. Copper nanoparticles have been used to enhance the physical and chemical properties of various dental materials, such as dental amalgam, restorative cements, adhesives, resins, endodontic-irrigation solutions, obturation materials, dental implants, and orthodontic archwires and brackets. The objective of this review is to provide an overview of copper nanoparticles and their applications in dentistry.

## 1. Introduction

Copper is a popular element in medical and dental research due to its antimicrobial properties and low toxicity [1,2]. The antimicrobial activities are induced by either copper’s metal ions or the oxidized cupric ions derived from copper nanoparticles (1–100 nm). Moreover, copper is readily available for the synthesis of copper nanoparticles, so it is cost effective [3]. Copper nanoparticles can be processed either naturally or via chemical synthesis [4,5,6,7]. In addition, they can easily oxidize in air or water and produce copper oxide nanoparticles. Like most metal nanoparticles that are commonly used in dentistry, copper particles have variable nano-sizes and forms, a unique distribution, and a large surface-area-to-volume ratio. These properties enhance the bio-physio-chemical functionalization, antimicrobial activity, and biocompatibility of the nanoparticles. Reports have shown that copper oxide nanoparticles are antimicrobial and inhibit biofilm formation [8,9]. The high surface-area-to-volume ratio of copper nanoparticles enhances their antimicrobial ability [10]. These nanoparticles’ antibacterial activities have been widely investigated, although the exact mechanism of copper nanoparticles against microbes is not clear [11,12,13]. Copper nanoparticles have demonstrated higher bactericidal activity against *E. coli, B. subtilis,* and *S. aureus* compared with silver nanoparticles, which are one of the nanoparticles commonly used nanoparticles in biomedical research [14,15].

Researchers want to develop dental materials that have antimicrobial properties to prevent oral infection. Copper nanoparticles exhibit antimicrobial activities and other metallic properties associated with dental applications. Preparing these nanoparticle composites with existing dental materials is easy and is said to be physiochemically stable. Still, they have a very limited clinical application. In dentistry, copper nanoparticles have mostly been studied as a modifier in amalgam and antimicrobial agents. Recently, several dental materials were studied with regard to copper nanoparticles, and it was reported that copper nanoparticles can be added to dental cements, restorative materials, adhesives, resins, irrigating solutions, obturations, orthodontic archwires and brackets, implant surface coatings, and the bone regeneration process [15,16,17,18,19,20]. The results were impressive but a bit ambiguous for clinical applications.

A search in PubMed using ((copper nanoparticles) AND (dentistry OR dental)) found no relevant review articles. The aim of this review is to highlight the research directions, their outcomes, and the feasibility of future studies. Therefore, we performed a systematic search of the publications in English and included the maximum number of works in the literature. We searched three common databases, namely, EMBASE, Google Scholar, and MEDLINE. In the search, the keywords were (copper OR (copper nanoparticles) OR (copper nanocomposite)) AND (dentistry OR (dental material)). These keywords covered as much information about copper nanoparticles in dentistry as possible without overlooking related research. This review includes all publications on the application of copper nanoparticles in dentistry. Abstracts, editorials, and letters to the editor were excluded. Among the vast number of articles reviewed, we have summarized the dental materials with copper nanoparticles into four main categories. They are metals and alloys, polymers and resins, restorative cements, and miscellaneous dental materials.

## 2. Antibacterial Mechanism of Copper Nanoparticles

Copper is a well-established antimicrobial and anti-inflammatory agent with a long history of medicinal applications [21,22]. The nano-sized particles and high surface-area-to-volume ratio allow copper nanoparticles to exhibit broad-spectrum antibacterial and antiviral activities [12]. Copper nanoparticles may have a similar mode of action as other metal nanoparticles. Although several studies have shown that copper-nanoparticle-containing materials demonstrate antibacterial activity and biocompatibility, related antibacterial mechanisms are not inclusive [23]. A few studies were conducted in an attempt to explore the antibacterial mechanism of copper nanoparticles. Mainly, three hypothetical mechanisms were commonly described. First, copper nanoparticles accumulate in the bacterial membrane and change its permeability. They then remove membrane proteins, lipopolysaccharides, and intracellular biomolecules and cause the dissipation of the proton-motive energy around the plasma membrane [24,25,26]. Second, reactive oxygen species from nanoparticles (in the form of nanoparticles or ions) process post-oxidative damage in cellular structures [27,28,29,30]. Third, cells’ uptake of ions (generated via nanoparticles) decreases intracellular adenosine triphosphate production and deoxyribonucleic acid (DNA) replication [31,32,33,34].

Studies demonstrated that the carboxyl group of bacterial lipoproteins has a negative charge that attracts positive copper ions. After binding with copper ions, bacterial cell membranes and their permeability change, and copper ions enter the cells. When they merge with phosphorus- and sulfur-containing biomolecules (i.e., DNA), the copper ions alter the cell structures and cell proteins. This alteration inhibits the cell’s biochemical processes and causes cell death [35,36]. However, copper ions inhibit enzymatic activity. They alter DNA or protein synthesis, inactivate their enzymes, and promote hydrogen peroxide production [37]. In addition, nanoparticles denature protein molecules by interacting with their sulfhydryl group [37]. At the same time, DNA, ribonucleic acid, proteins, and cytoplasm leak out through the permeable membrane, causing cell damage that kills bacteria [38]. Furthermore, several other hypotheses of possible modes of copper nanoparticle and bacterial interaction exist, such as the permeabilization of the plasma membrane, the peroxidation of membrane lipids, changes in proteins, the inhibition of protein assembly and activity, and the deformation of nucleic acids [39]. In most cases, all of these are directed in the same way. We believe that further in vitro and in vivo studies should be conducted to produce a systematic outline [40].

Many studies investigated copper nanoparticles’ antibacterial activities. However, few studies reported their antiviral action. They demonstrated that copper nanoparticles target the viral genome, especially encoding the genes responsible for viral infections [21,41]. In addition, some studies showed a similar reactive oxygen species process in the viral envelope or capsid, which resembles antibacterial activity [21]. Viruses are more vulnerable to injuries induced by copper nanoparticles because, unlike bacteria and fungi, they do not have a repair mechanism. This leads to instant cell death [39]. Processes that cause the immediate inactivation of microbes upon contact are known as “contact killing” [42]. In many studies, researchers have taken advantage of this “contact killing” property and have created copper-nanoparticle-functionalized antiviral surfaces. Copper nanoparticles are co-integrated into surface research to increase “contact killing”, as well as to develop antibacterial and antiviral combination effects [43]. Figure 1 demonstrates the mechanism of copper nanoparticles’ antibacterial activities in bacterial cells.

Attach to the cell wall and release copper ions.Adhere to the cell membrane and enter the cell.Disrupt the cell wall and membrane, denature protein, interrupt enzymatic activity, interrupt deoxyribonucleic acid (DNA) replication, and interrupt adenosine triphosphate (ATP) production.Generate reactive oxygen species. Reactive oxygen species further interrupt DNA replication and initiate the breakdown of DNA, denature ribosomes, and denature protein.

## 3. Copper Nanoparticles in Dental Materials

Copper nanoparticles present antimicrobial and bio-physio-chemical properties. They enrich the material pool to minimize the shortage of dental materials in various clinical applications. Generally, copper nanoparticles are used in dental metals and alloys, dental polymers and resins, dental cements, and miscellaneous dental materials (Table 1).

### 3.1. Copper Nanoparticles in Dental Metals and Alloys

Prosthesis-induced inflammatory diseases, such as stomatitis and peri-implantitis, pose challenges in clinical dentistry. Several strategies have been applied to solve these problems. However, no specific solution has yet been found. Copper nanoparticles may play a role in controlling infections. Copper nanoparticles destroy genomic and plasmid DNA, making them an ideal alternative to antimicrobial surface coatings [44]. In addition, copper nanoparticles’ “contact killing” ability may also be well considered. Copper-nanoparticle-incorporated removable and fixed partial denture framework designs could solve denture-induced stomatitis and oral infections [45].

Dental amalgam restoration occupies a unique position in dentistry. These amalgam alloys are broadly known as low-copper (5% or less copper) and high-copper alloys (13% to 30% copper). A study reported that through the increasing of their copper density, conventional dental amalgam alloys improved their microstructural and mechanical properties. Reports have also revealed the disappearance of the gamma-2 phase in copper content with more than 20 wt% [46]. High-copper amalgams remain the leading materials in Europe, America, and other advanced markets. A report stated that non-gamma-2 amalgams had superior strength and corrosion resistance properties [47]. In addition, high-copper amalgam alloys showed less marginal deterioration in clinical studies, resulting in durable restorations [48]. Studies revealed that through the increasing of copper nanoparticles in amalgam, the gamma-2 phase can be eliminated to increase the amalgam’s compressive strength [49].

Implant dentistry has always faced the problem of peri-implantitis. Microbial infection is one of the most common postoperative complications of implant surgery to be resolved. It is a complex situation, and, apparently, eliminating its occurrence is the only way to overcome it. However, it is generally impossible to stop the incidence completely. The advanced biomaterial research confirmed that biofilm formation on the implant surface is the main cause, and it leads to the serious consequences of implant-related infections [112,113]. In recent decades, studies focused on developing new implant materials or applying various coatings on existing implants using nanoparticles to inhibit biofilm formation and to prevent biofilm-related infections. Some studies involved applying antibiotic surface coatings to metal implants to develop prolonged antimicrobial action [114,115]. However, this process was not successful due to the unstable bond between the surface coatings and implants, as well as the risk of antibiotic resistance due to the continuous release of antibiotics in the body [116,117]. Many researchers have used copper nanoparticles for the surface modification of metallic implants [118,119].

Nisshin Steel (Tokyo, Japan) made the first copper-linked antimicrobial stainless steel for biomedical applications [113]. Several studies have reported using copper nanoparticles as an antimicrobial surface coating on different types of medical-grade stainless steel, such as 317L stainless steel [23,50,51,52,53,113], 316L stainless steel [23,54], 304 stainless steel [55], and 420-copper stainless steel [56]. Some studies have reported using copper nanoparticles for the development of a magnesium–copper alloy as an implant material [57,58]. Several studies have also reported using copper nanoparticles for the surface modification of titanium implants [57,58,59,60,61,62,63]. Moreover, some studies have shown that the titanium–copper alloy exhibits antibacterial and anti-aging properties. The studies have also demonstrated that these antimicrobial properties could be tuned via changing the copper concentration of the alloy composition [64,65,66]. Another study stated that copper-containing mesoporous bio-glass reduced bacterial activity and biofilm formation by releasing copper ions [67,68,69].

A study reported that copper nanoparticles coating dental implant healing caps inhibited bacteria and biofilm formation [70]. On the other hand, another report showed that a copper-bearing titanium alloy implant exhibited anti-infective properties against oral bacteria. They also demonstrated that a titanium–copper alloy not only inhibited peri-implant infections but also possessed biocompatibility [71]. Copper-nanoparticle hydroxyapatite is antibacterial. A study reported that a titanium–copper alloy and a titanium-copper ion-doped hydroxyapatite inhibited oral bacteria [72]. A Titanium-copper alloy reduced the formation of biofilms to prevent implant failure. It was also effective against implantitis-associated oral bacterial species. Therefore, the study recommended using a titanium–copper alloy as an alternative for dental implants [61]. These are still in the laboratory stage, and, therefore, further clinical studies should be aimed at validating the application of copper-nanoparticle-functionalized implants in clinical implant dentistry.

The inclusion of copper nanoparticles in a nickel–titanium alloy offers several benefits in orthodontic appliances. Copper nanoparticles in archwire reduced loading stress and provided a relatively high unloading stress, which increased orthodontic tooth movement and was explained as a lower stress hysteresis [73,74]. A study demonstrated that the addition of copper nanoparticles to metallic orthodontic appliances developed clinical significance in terms of hyperelasticity, conversion temperature, and load cycling behavior [73]. Studies have furthermore reported that the presence of copper nanoparticles in the nickel–titanium orthodontic archwire reduced the aging effect and galvanic corrosion [73,75].

Studies showed that copper–nickel–titanium wires improved mechanical and thermal properties [76,77]. The addition of copper nanoparticles increased friction under both wet and dry conditions [78]. Furthermore, a clinical study compared copper–nickel–titanium archwire and nickel–titanium archwire for correcting mandibular incisor crowding and found no impact on the correction of the crowding [79]. Another study found that the benefits of the loading pattern of the copper–nickel–titanium wire obtained from the laboratory were not reflected in the clinical settings. Therefore, further investigations were suggested to ensure and control these wires’ clinical performance in orthodontic practice.

### 3.2. Copper Nanoparticles in Dental Polymers and Resins

A heat-cured thermoset denture base with copper oxide nanoparticles is effective to inhibit the growth of *C. albicans,* which is mainly responsible for denture stomatitis [120]. Moreover, another study found that copper oxide nanoparticles released ions while they were incorporated into different resin formulations (i.e., heat-cured acrylic resin denture, chemically cured soft liner, and cream-type adhesive). This ion release could be controlled and utilized in therapeutic drug delivery for the treatment of oral diseases. However, the release of these ions may vary. A study showed that the denture base and adhesive had higher releases compared with the denture liner [80]. Researchers may explore the controlled and optimum therapeutic release of copper ions for an optimum clinical outcome. A study reported copper-doped mesoporous bioactive glass nanosphere-incorporated resin exhibited antimicrobial properties, mechanical properties, and a better aging resistance effect [81]. An etch-and-rinse adhesive with copper nanoparticles appeared to have antimicrobial activity and prevented the degradation of the adhesive interface without altering the mechanical properties [82]. Polyacrylic acid-coated copper iodide nanoparticles in another study acted as an antibacterial additive to adhesive, and their bonding strength or biocompatibility was not affected [83]. Another study showed that copper nanoparticles added to an adhesive improved the shear bond strength and antibacterial properties without cytotoxicity [121].

### 3.3. Copper Nanoparticles in Dental Cements

Several in vitro studies showed that copper cements developed compressive strength, solubility, and antibacterial activity, and they were recommended for use as cariostatic lining under a less soluble restorative material [84,85,86]. Some studies also revealed that copper in restorative materials reduced microorganisms’ growth and viability [122] and improved the bond on the teeth interface [123]. Moreover, fluoridated amalgam demonstrated better caries prevention in both primary and secondary caries adjacent to the restoration [124]. A study reported photo-polymerized copper(I)-catalyzed azide-alkyne cycloaddition composites to be mechanically strong and highly tough materials. In addition, they reduced shrinkage stress and generated a modest exothermic reaction [87]. Another study showed that copper nanoparticles added to a commercial glass ionomer developed antibacterial activity against oral strains [88]. On the other hand, blue calcium phosphate cement with copper ions in another study showed a synergized dual antibacterial effect and cytocompatibility [89]. Many studies have shown that calcium phosphate cement with copper phosphate nanoparticles could promote vascularized new bone formation around cancerous bone defects [90,91]. On the other hand, a copper-modified zinc oxide phosphate cement showed low surface allocations of copper but no improvement in its antimicrobial properties [92]. Therefore, with this limited study, it is difficult to justify the clinical application. In light of this, the optimization of copper use in restorative materials for clinical applications should be investigated further in relation to the various strains associated with prosthesis-induced oral infection.

### 3.4. Copper Nanoparticles in Miscellaneous Dental Materials

Copper is the third-most abundant trace element in the human body [125,126]. Some reports stated that the lack of copper had a negative influence on immune cells (i.e., neutrophils, macrophages, T cells, and natural killer cells), decreased interleukin-2 production, and alternately enhanced proinflammatory cytokine production (e.g., tumor necrosis factor-α, and matrix metalloproteinase-2 and -9), which degraded the collagen and extracellular matrix components in the periodontal ligament [93]. Because copper acts as a cofactor for metalloenzyme (i.e., superoxide dismutase), which is an essential antioxidant for chronic periodontitis, an optimal level of copper is essential for the prevention of inflammatory passages [93].

Copper is essential for the development of connective tissue. The elevation of serum copper has reflected the changes occurring in the periodontal collagen metabolism of periodontitis patients [94]. A study reported an improvement in chronic periodontitis in diabetic and non-diabetic patients using increased copper ions levels at baseline for nonsurgical periodontal therapy [95]. Several studies have also reported increased copper ion levels in patients suffering from chronic and aggressive periodontitis, as well as in acute and chronic gingivitis [96,97,98]. A study reported a link between the salivary copper ion level and periodontitis [99]. However, the exact mechanism of the increase in copper in the plasma and its association with periodontal disease is still unclear and needs to be identified. Some studies showed the therapeutic application of copper nanoparticles for periodontal therapy. They demonstrated that copper-nanoparticle-based antimicrobial ions inhibited bacterial growth. They also observed the controlled and sustained release of bactericidal copper concentrations for localized periodontal therapy [100]. Furthermore, some studies used copper-nanoparticle-containing composite materials for periodontal therapy. They used a sodium copper chlorophyllin solution to reduce the production of volatile sulfur compounds by inhibiting the periodontal anaerobes associated with malodour, and they recommended using it to improve oral and periodontal health [101].

Another study used copper-calcium hydroxide nanoparticles for treating apical periodontitis in an endodontically treated tooth, and it recommended using them to treat periodontitis [102]. Researchers have used several copper nanoparticle formulations for periodontal therapy. However, these nanoparticle formulations are still limited in laboratory research, and their effects in clinical applications remain unclear. Further dedicated research may improve the use of copper nanoparticles in antimicrobial, anti-inflammatory, and regenerative periodontal therapy.

Endodontic treatment mostly fails due to microbial infections. The inadequate disinfection of root canals is a main cause of post-treatment reinfection [127]. Studies showed that bacteria can survive inside the root canal after a careful chemo-mechanical preparation [128]. Copper nanoparticles, due to their ionic compounds, have demonstrated the potential to produce and capture electrons, as well as to generate radical oxygen species. These particles have led to toxic hydroxyl radical production and constitute an antimicrobial agent in endodontic treatment [103,104]. On the other hand, copper nanowires have shown excellent antimicrobial effects against the oral microbes associated with the endodontic systems, so it is suggested that they be considered for root canal disinfection [105]. Moreover, a study explained the antibacterial effect of copper sulfate nanoparticles and proposed further clinical trials for clinical settings [106]. Biofilm is resistant to common intracanal irrigation, antimicrobial drugs, and the host immune response. A study reported that in such cases, copper hydroxide nanoparticle–based endodontic paste can reduce the growth and replication time of root canal system–associated oral microbes, which affects the formation and persistence of biofilm. The researchers suggested using copper hydroxide nanoparticle–based endodontic paste for the prevention and treatment of biofilm-associated endodontic infections [107].

With the progress of multifunctional biomaterials in bone grafting, the possibility of osteogenesis and angiogenesis has attracted attention in bone and tissue regeneration research. Inspired by the antibacterial activity of copper ions, a study reported developing a copper-nanoparticle-based nonstoichiometric dicalcium silicate for infectious bone repair [108]. Another study reported that a copper-doped biphasic calcium phosphate powder that was made of hydroxyapatite and β-tricalcium phosphate powder exhibited antibacterial activity. In addition, they reported that it had good adherence to bone marrow cells and maintained good cell viability. Therefore, they recommended it as a promising bioceramic for bone substitution and/or prosthesis coatings [109,110]. A study reported using graphene oxide copper nanocomposites for bone regeneration and revealed vascularized new bone formation [111]. Although studies are limited in this field, they are promising. Clinical studies using copper nanoparticles in various tissue engineering strategies may result in a new direction of regenerative biomaterials.

## 4. Biocompatibility Studies of Copper Nanoparticles

The biocompatibility of nanoparticles is an important concern these days. The rapid growth of nanotechnology is rampant in every research field, including biomedical research. However, to date, no authentic information exists on the inferences of nanoparticles on human health [129]. Copper is a necessary trace element, and its deficiency is conducive to various diseases in humans. In addition, it acts as a catalyst cofactor in some redox enzymes that are essential for broad-spectrum metabolic processes. On the other hand, if the copper intake exceeds the tolerable limit, it shows toxic effects that lead to cell death [130]. Despite the great potential of the biomedical application of copper nanoparticles, toxicity studies of these nanoparticles are mostly confined to in vitro studies.

Copper nanoparticles have shown toxic effects on several cell lines [131,132,133,134,135,136,137]. Several in vitro studies have been conducted. However, only a few studies reported the in vivo toxicity of copper nanoparticles [138,139,140,141]. No information was provided on the bioavailability and excretion data of long-term exposure to copper nanoparticles. Future research is needed to provide a detailed and systematic overview of both the in vitro and the in vivo toxicity of copper nanoparticles, as well as their kinetics. Recently, the preliminary investigation of the biocompatibility of copper nanoparticles showed that they have toxicity in both humans and the environment [141,142]. Even though copper is sustained in the homeostasis of the human body, excess copper showed toxic effects on the kidney and liver [142,143]. Although the possible risks of copper nanoparticles have been identified in human health, their subacute toxicity has not yet been defined.

Copper nanoparticles that are 23.5 nm in size are considered to be class 3 medium toxic materials, with an LD50 value of 413 mg/kg of body weight. Copper nanoparticles’ toxicity is related to their ionization [138]. Copper nanoparticles lead to ultra-high reactiveness due to their large surface and active functional ions. They react with hydrogen ions in the gastric juice and produce a large amount of hydrogen carbonate, and their excretion leads to kidney disorders. In addition, a study showed that the toxicity of copper nanoparticles depends on gender. Male mice showed more severe toxic symptoms than females after being exposed to the same mass of nanoparticles [138]. Another study also identified that the increased production of reactive oxygen species and reactive nitrogen species plays an important role in copper nanoparticle-induced organ impairment [144].

The interaction and impact of nanoparticles on cells and tissues have been explained differently based on their distribution, particle size, and penetration capacity. Studies have also reported that the variation of the effects depends on the different synthesis methods of nanoparticles [145]. Therefore, with this limited study, it is difficult to bring a gross scenario. The effect and involvement of nanoparticles should be clearly defined. Thus, detailed studies focusing on their toxicity, bioavailability, kinetics, and biodistribution in different organs, including the liver, brain, lung, heart, and spleen, should be further investigated. Although, some studies described the potentials of copper and other metallic nanoparticles in specific or different aspects of dental applications, the biosafety was not clearly identified [146,147,148]. At the same time, the synthesis, the morphological and physicochemical properties of copper nanoparticles, and these properties’ impacts also need to be explored. Hence, concrete information regarding copper nanoparticles’ biocompatibility will be in the frame for their future applications.

## 5. Conclusions

Copper nanoparticles play a dual role in the development of the properties of dental materials. The inclusion of copper nanoparticles may improve the physio-mechanical properties and introduce or enhance the antimicrobial activities of various dental materials. It is expected that researchers and clinicians will focus on the perspective of cost-effective copper nanoparticles in dentistry. This will reveal potentials and limitations, as well as open a new door to dental biomaterials research for the use of copper nanoparticles in clinical dental practice.

## Figures and Tables

**Figure 1 nanomaterials-12-00805-f001:**
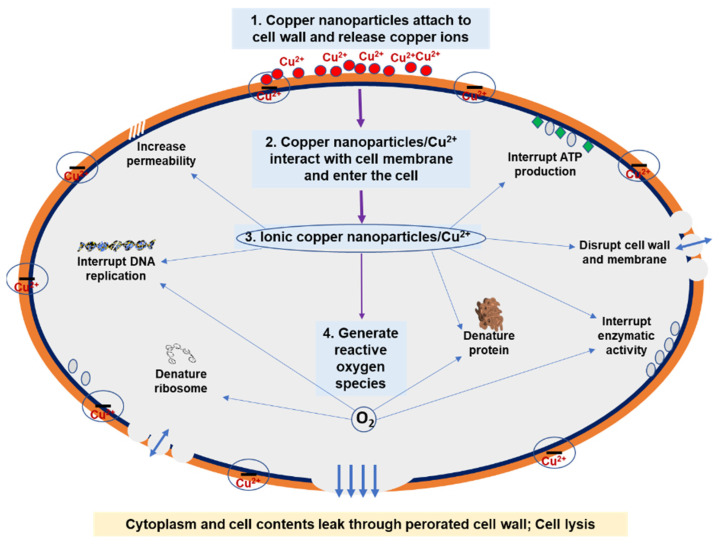
Antibacterial mechanism of copper nanoparticles.

**Table 1 nanomaterials-12-00805-t001:** Properties, applications, and functions of dental materials with copper nanoparticles.

Materials [Reference(s)]	Properties	Applications	Functions
** *Dental metals and alloys* **			
Copper-coated metal [44,45]	Offer antimicrobial properties	Denture framework	Prevent stomatitis
Copper amalgam alloy [46,47,48,49]	Improve microstructural Improve mechanical properties	Amalgam restoration	Prevent corrosion
Copper-linked alloy [50,51,52,53,54,55,56]	Offer antimicrobial properties	Dental implant	Prevent implantitis
Magnesium–copper alloy [57,58]	Offer antimicrobial properties	Dental implant	Prevent implantitis
Titanium–copper alloy [57,58,59,60,61,62,63,64,65,66,67,68,69,70,71]	Offer antimicrobial properties	Dental implant	Prevent implantitis
Titanium–hydroxyapatite–copper alloy [72]	Offer antimicrobial properties	Dental Implant	Prevent implantitis
Nickel–titanium–copper alloy [73,74,75,76,77,78,79]	Enhance mechanical properties Enhance thermal properties Prevent galvanic corrosion Reduce alloy aging	Orthodontic bracket Orthodontic archwire	Facilitate orthodontic tooth movement
** *Dental polymers and resins* **			
Copper-nanoparticle-incorporated acrylic resin [80]	Offer antimicrobial properties	Denture soft liner	Prevent stomatitis
Copper-doped mesoporous bioactive glass nanosphere acrylic resin [81]	Facilitate copper-ion release Offer antimicrobial properties	Denture acrylic base	Prevent stomatitis
Copper nanoparticles with adhesive resin [82]	Offer antimicrobial properties	Dental adhesive	Prevent secondary caries
Polyacrylic acid–copper iodide nanoparticles with adhesive resin [83]	Offer antimicrobial properties	Dental adhesive	Prevent secondary caries
** *Dental Cements* **			
Copper [84,85,86]	Offer antimicrobial properties	Lining materials	Prevent secondary caries
Copper(I)-catalysed azide-alkyne cycloaddition composites [87]	Reduce shrinkage stress	Composite resin restoration	Prevent secondary caries
Copper nanoparticles incorporated in glass ionomer cement [88]	Offer antimicrobial properties	Glass ionomer restoration	Prevent secondary caries
Copper ions releasing blue calcium phosphate cement [89]	Offer antimicrobial properties Improve cytocompatibility	Regenerative dental material	Promote bone formation
Functionalized copper phosphate nanoparticles [90,91]	Increase vascularization Enhance bone regeneration	Regenerative dental material	Promote bone formation
Copper-modified zinc oxide phosphate [92]	Reduce marginal gap	Luting cement	Prevent secondary caries
** *Miscellaneous* ** ** *dental materials* **			
Copper nanoparticles [93,94,95,96,97,98,99,100,101,102]	Enhance anti-inflammatory effects	Periodontal therapy	Prevent inflammation
Copper-based substance [103,104,105,106,107]	Offer antimicrobial properties	Endodontic irrigation solution Endodontic paste	Prevent apical reinfection
Nano copper-nonstoichiometric dicalcium silicate [108]	Offer antimicrobial properties Facilitate tissue regeneration	Regenerative dental material	Promote bone formation
Copper-doped biphasic calcium phosphate [109,110]	Improve bone regeneration Act as a bone substitution	Synthetic bone graft material	Promote bone formation
Graphene oxide copper nanocomposite [111]	Enhance bone regeneration	Regenerative dental material	Promote bone formation

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
