# Peer review of "Application of Copper Nanoparticles in Dentistry"

_nanomaterials, 2022, doi:10.3390/nano12050805_

Round 1

Reviewer 1 Report

The abstract says:  "Metals nanoparticles are commonly used in dentistry for their exclusive shape-dependent properties, including their variable nano sizes and forms, unique distribution, and large surface area to volume ratio. These properties enhance the bio-physio-chemical functionalization, antimicrobial activity, and biocompatibility of the nanoparticles."  However, the article does not approach the influence of the shape and size of these particles, only the antimicrobial activity. Also, the article does suggest how to sinterized this particles and the consequences that the sinterized method may cause in the bio-physio-chemical functionalization of the particle. 

I suggest an iten about:

  1. Sinterized methods
  2. influence of shape and size of nanoparticles

Reviewer 2 Report

Number of studies including narrative reviews are available about copper particles and their use, properties and antimicrobial properties. 

Narrative review does not add much to the evidence. It will be a considerable contribution to the scientific knowledge, if authors can revise it to be a systematic review with meta-analysis.

Author Response

Dear Editor,

Thank you for reviewing our manuscripts for consideration for publication in Nanomaterials. While we are encouraged by the comments of the first and third reviewers, we are disappointed with the subjective comments by reviewer 2. We attached a point to point response as below for your perusal.

Reviewer 2's comments

Authors’ point to point response

1) Number of studies including narrative reviews are available about copper particles and their use, properties and antimicrobial properties.

We disagree with this comment. A search in PubMed using [(copper nanoparticles) AND (dentistry OR dental)] found no relevant review articles. We added this in line 58 highlighted in yellow.

Moreover, this paper gives an overview of copper nanoparticles, not "copper particles".

2) Narrative review does not add much to the evidence. It will be a considerable contribution to the scientific knowledge,

     if authors can revise it to be a systematic review with meta-analysis.

We disagree with this comment.

This review is probably the first review on copper nanoparticles for dental use. Our review reported application of copper nanoparticles in dental metals and alloys, dental polymers and resins, dental cements, and miscellaneous dental materials.

Meta-analysis is the use of statistical methods to summarize the results of studies, such as effectiveness of an intervention in clinical trials. It cannot be used in this type of review.

The reviewer should pay attention to our paper’s objective, which is to provide an overview of copper nanoparticles and their applications in dentistry. We systematically searched the literature and 22 identified materials. We grouped these materials into 4 categories for reporting in our manuscript, which is probably the first review on copper nanoparticles for dental use. Because of the subjective comments, we request to not to invite Reviewer 2 to continue  reviewing our paper.

Thank you for your attention,

C H

Professor Chun-Hung Chu

Associate Dean and Professor,

Faculty of Dentistry, The University of Hong Kong

Reviewer 3 Report

This is an interesting review covering all the "copper use" areas in the broad spectrum of dental applications. I detected minor grammatical and syntax errors which have been commented on in the revised pdf file. However, I have a comment: how did the authors ensure that they have included most of the published articles? Did they use specific keywords on different databases? Although this study was not a systematic review, it should contain a description of how researchers selected the included studies.
